# High-resolution Mapping of Nitrogen Oxide Emissions in Large US Cities from TROPOMI Retrievals of Tropospheric Nitrogen Dioxide Columns

Fei Liu[1,2], Steffen Beirle[3], Joanna Joiner[2], Sungyeon Choi[2,4], Zhining Tao[1,2], K. Emma Knowland[1,2], Steven J. Smith[5], Daniel Q. Tong[6,7], Siqi Ma[6,7], Zachary T. Fasnacht[2,4], Thomas Wagner[3]

[1]Goddard Earth Sciences Technology and Research (GESTAR) II, Morgan State University, Baltimore, MD 21251, USA
[2]NASA Goddard Space Flight Center, Greenbelt, MD, 20771, USA
[3]Max-Planck-Institut für Chemie, Mainz, 55128, Germany
[4]Science Systems and Applications Inc., Lanham, MD, 20706, USA
[5]Joint Global Change Research Institute, Pacific Northwest National Laboratory, College Park, MD, 20740, USA
[6]Department of Atmospheric, Oceanic and Earth Sciences, George Mason University, Fairfax, 22030, Virginia, USA
[7]Center for Spatial Information Science and Systems, George Mason University, Fairfax, 22030, Virginia, USA

*Correspondence to*: Fei Liu (fei.liu@nasa.gov)

**Abstract.** Satellite-derived spatiotemporal patterns of nitrogen oxide ($NO_x$) emissions can improve accuracy of emission inventories to better support air quality and climate research and policy studies. In this study, we develop a new method by coupling the chemical transport Model-Independent SATellite-derived Emission estimation Algorithm for Mixed-sources (MISATEAM) with a divergence method to map high-resolution $NO_x$ emissions across US cities using TROPOspheric Monitoring Instrument (TROPOMI) tropospheric nitrogen dioxide ($NO_2$) retrievals. The accuracy of the coupled method is validated through application to synthetic $NO_2$ observations from the NASA-Unified Weather Research and Forecasting (NU-WRF) model, with a horizontal spatial resolution of 4 km × 4 km for 33 large and mid-size US cities. Validation reveals excellent agreement between inferred and NU-WRF-provided emission magnitudes (R = 0.99, Normalized Mean Bias, NMB = -0.01) and a consistent spatial pattern when comparing emissions for individual grid cells (R = 0.88 ± 0.06). We then develop a TROPOMI-based database reporting annual emissions for 39 US cities at a horizontal spatial resolution of 0.05°×0.05° from 2018 to 2021. This database demonstrates a strong correlation (R = 0.90) with the national emission inventory (NEI) but reveals some bias (NMB = -0.24). There are noticeable differences in the spatial patterns of emissions in some cities. Our analysis suggests that uncertainties in TROPOMI-based emissions and potential misallocation of emissions and/or missing sources in bottom-up emission inventories both contribute to these differences.

## 1 Introduction

Nitrogen oxides ($NO_x$), including nitrogen dioxide ($NO_2$) and nitric oxide (NO), play a vital role as trace gases in the atmosphere. They substantially contribute to the formation of secondary aerosols and tropospheric ozone (Seinfeld and Pandis, 2006), which in turn impact climate and human health. The primary source of $NO_x$ emissions is fossil fuel combustion from mobile and industrial sources, often concentrated in urban areas (Crippa et al., 2018). Traditionally, $NO_x$ emissions are compiled using "bottom-up" methods that rely on total fuel usage and average emission factors. However, determining urban emissions is challenging due to the difficulty of obtaining comprehensive data on operating conditions of emitters and their rapidly changing emission factors (Liu et al., 2016b) at the city level (Butler et al., 2008). Spatial distribution of emissions is commonly estimated by applying proxies, such as road network maps and/or population density maps, to allocate total emissions onto a grid. This process may introduce significant biases due to spatial mismatches between emission locations and spatial proxies (Woodard et al., 2014; Hogue et al., 2016).

Tropospheric $NO_2$ vertical column densities (VCDs) retrieved from satellite observations offer valuable insights into the magnitude and location of global $NO_x$ emissions (Martin et al., 2003; Lamsal et al., 2011). Techniques leveraging Chemical Transport Models (CTMs) have been developed to relate $NO_2$ VCDs to $NO_x$ emissions. For example, the extended (Ding et al., 2017) and ensemble (Miyazaki et al., 2017) Kalman filter, the four-dimensional variational (4D-Var) method (Henze et al., 2007, 2009), and the hybrid mass balance/4D-Var (Qu et al., 2019).

Alternative methods, which are independent of CTMs, have also been proposed to characterize $NO_x$ plumes from major sources (e.g., Beirle et al., 2011; Liu et al., 2016a; Laughner and Cohen, 2019). Early studies employed one dimensional (1D) empirical plume dispersion functions to fit $NO_2$ VCDs surrounding isolated sources. Liu et al. (2016a, 2022) refined these functions to accommodate the description of $NO_2$ plumes from sources in polluted background. The derived fitting parameters yield the magnitude of $NO_x$ emissions for point sources, such as power plants (de Foy et al., 2014), and cities, by assuming these sources as equivalent point sources (Lu et al., 2015; Liu et al., 2017; Goldberg et al., 2019). A recent study (Beirle et al., 2019) introduced a two-dimensional (2D) divergence approach, enabling the identification of finer details in $NO_x$ distributions and thus facilitating the detection of smaller sources. Subsequent studies have further refined this approach to enhance divergence calculation (de Foy and Schauer, 2022) and to optimize its performance over mountainous regions (Sun, 2022). Current applications of this divergence method primarily focus on inferring emissions from point sources, e.g., power plants in South Asia (de Foy and Schauer, 2022), oil and gas production areas in the United States (Dix et al., 2022), and a global catalog of point sources (Beirle et al., 2021, 2023). However, the robustness of applying the approach for area sources such as cities has seldom been investigated.

The main goal of this study is to map $NO_x$ emissions across major cities in the US based on the TROPOspheric Monitoring Instrument (TROPOMI; Veefkind et al., 2012) retrievals of $NO_2$ VCDs (Ialongo et al., 2020). We will couple the 1D (Liu et al., 2022) and 2D (Beirle et al., 2019) CTM-independent approaches to infer gridded $NO_x$ emissions. Due to the absence of established "true emissions" to serve as a standard for comparison, gauging the precision of the coupled method becomes

challenging. We will employ synthetic $NO_2$ observations, generated by a model simulation, to appraise the accuracy of the approach. Section 2 offers a summary of the satellite data used to map urban $NO_x$ emissions, and the synthetic $NO_2$ observations used for validation. In Section 3.1, we assess the coupled approach by juxtaposing the emissions inferred from synthetic $NO_2$ observations with the true emissions used by the model for generating the synthetic observations. In Section 3.2, we compare the emissions derived from satellite data with the National Emission Inventory (NEI) developed by the United States Environmental Protection Agency (US EPA) to shed light on the uncertainties of both and bottom-up emissions. Section 3.3 summarizes the uncertainties of the method and the TROPOMI-derived emissions. We discuss the robustness of the derived emissions and outline the plan work in Section 4.

## 2 Data and Methods

### 2.1 TROPOMI $NO_2$ dataset

TROPOMI is a UV-VIS-NIR-SWIR nadir-viewing imaging spectrometer (Veefkind et al., 2012) on board the Sentinel 5 Precursor (S5P) satellite that was launched in 2017. It has a ground pixel size at nadir of 7.5 km×3.5 km before August 6, 2019, and improved to 5.5 km ×3.5 km afterwards. It provides daily global coverage with a local equator crossing time of approximately 13:30 h. The instrument's radiance and irradiance measurements are utilized to obtain slant $NO_2$ columns using the Differential Optical Absorption Spectroscopy (DOAS) algorithm (Platt and Stutz, 2008). The slant columns are subsequently differentiated into stratospheric and tropospheric parts, with the tropospheric slant columns being further converted to vertical columns based on air mass factors.

We selected TROPOMI $NO_2$ retrieved by NASA Goddard Space Flight Center (GSFC), TROPOMI Multi-Decadal Nitrogen Dioxide and Derived Products from Satellites (MINDS) $NO_2$ product (Lamsal et al., 2022), in this study due to its utility in investigating emission trends. The official $NO_2$ product available at the commencement of this study switched the processor version in December 2020, which introduced a discontinuity in the time series (van Geffen et al., 2022). The GSFC product (Lamsal et al., 2022) utilizes slant columns from the official product to retrieve a full time series of tropospheric $NO_2$ VCDs from spring to autumn months of 2018 to 2021 based on the algorithm developed for the Ozone Monitoring Instrument (OMI) Aura $NO_2$ standard product version 4.0 (Lamsal et al., 2021). TROPOMI $NO_2$ data has been reported to have an overall low bias (Judd et al., 2020; Tack et al., 2021; Verhoelst et al., 2021; Wang et al., 2020), which will be propagated into emissions inferred from TROPOMI $NO_2$.

We further aggregate TROPOM's individual measurements at spatial resolution of 0.05°×0.05° by averaging the original pixels weighted by the fraction of the overlapping surface area. Only high-quality pixels with a quality assurance value (qa_value) above 0.75 are considered for averaging, which excludes cloud-covered scenes with cloud radiance fraction > 0.5. Following the approach of earlier research (e.g., Liu et al., 2017), our analysis is confined to the data from May through September. This

decision is made to omit winter data, characterized by longer NO$_x$ lifetimes, which in turn lead to greater uncertainties to the method.

## 2.2 Emission mapping algorithm

We couple our 1D CTM-Independent SATellite-derived Emission estimation Algorithm for Mixed-sources (MISATEAM; Liu et al., 2022) with the 2D divergency method of Beirle et al. (2019). We couple our 1D CTM-Independent SATellite-derived Emission estimation Algorithm for Mixed-sources (MISATEAM; Liu et al., 2022) with the 2D divergence method of Beirle et al. (2019). 1D MISATEAM quantifies the magnitude of city-level NO$_x$ lifetime and emissions by conceptualizing urban areas as point sources and thus does not capture the spatial variability within the urban areas. Conversely, the 2D divergence method allows for the resolution of finer spatial details in NO$_x$ distributions but relies on additional, often external, sources for determining NO$_x$ lifetimes, which can be a significant limitation. The coupled algorithm (hereafter referred to as 2D MISATEAM for simplicity) leverages the strengths of both: it maps NO$_x$ emissions over urban areas with enhanced spatial detail and does so independently by deriving lifetimes directly from NO$_2$ measurements, thereby overcoming the need for prescribed or externally sourced lifetime constraints. 1D and 2D MISATEAM methods deliver consistent estimates for total emissions, displaying a small relative difference of -5% ± 9%. While this result demonstrates the internal consistency of the MISATEAM methods, it's important to note that such coherence is due to the shared fundamental principle of mass balance underlying both methodologies. The cities listed in Table S1 that failed NU-WRF validation demonstrate a similar consistency between the two methods, with a relative difference 1% ± 14%. Therefore, the similarity in emissions estimations should not be viewed as an independent validation metric.

We average both NO$_2$ VCDs and reanalysis wind data from May to September each year. We then use those averaged data to infer NO$_x$ emissions $E$ by summing the divergence of the NO$_x$ flux $D$ with the NO$_x$ sink $S$ based on the continuity equation for steady state following:

$$E = D + S \tag{1}$$

$$D = R_{NO_x:NO_2} \times \nabla \cdot \vec{F} = R_{NO_x:NO_2} \times \nabla \cdot \vec{v}(\Omega - b) \tag{2}$$

$$S = R_{NO_x:NO_2} \times (\Omega - b)/\tau, \tag{3}$$

Where $\vec{F}$ is the NO$_x$ flux. It is calculated from the horizontal fluxes of NO$_2$ VCDs $\Omega$. $R_{NO_x:NO_2}$ is the ratio of NO$_x$ to NO$_2$ columns. Following previous studies (Beirle et al., 2019), we use an $R_{NO_x:NO_2}$ value of 1.32 to represent "typical urban conditions and noontime sun" (Seinfeld and Pandis, 2006). We interpolate the Goddard Earth Observing System Forward Processing for Instrument Teams (GEOS FP-IT) reanalysis wind vectors (Lucchesi, 2015) to the TROPOMI overpass time and average layers from surface to 1000 m altitude to derive $\vec{v}$ used in Eq. (2). Since the NO$_x$ sinks are dominated by the chemical loss through the reaction of NO$_2$ with OH at TROPOMI's local overpass time (13:30 local time), it can be characterized by a first order effective NO$_x$ lifetime $\tau$. Consequently, it bears a proportionate relationship to the NO$_2$ VCD

itself in Eq. (3). Note that we assume a relatively constant background and subtract the NO$_2$ background $b$ from $\Omega$ in the
125 calculation of the divergences and the sinks, because we aim to remove the natural and non-local contributions from the total emissions for each urban area in order to infer urban emissions. For each city, we infer $b$ and $\tau$ by applying 1D MISATEAM (Liu et al., 2022) to NO$_2$ VCDs averaged from May through September, 2018–2021, assuming $b$ and $\tau$ are constant over years. Additional technical details for deriving $b$ and $\tau$ are given in Fig. S1 and Text S1 of the Supplement.

The city of New York serves as a case study to showcase our approach. With its substantial size and numerous point and area
sources, this city is an ideal illustration of the capability of 2D MISATEAM to map emissions from various sources. Figure 1a–c illustrate maps of the derived sink $S$, the divergence $D$, and the resulting NO$_x$ emissions $E$, respectively. The divergence of the NO$_x$ flux (Fig. 1b) presents enhancement throughout the urban area due to emissions from traffic and industrial sources. The divergence is negative outside the urban areas, because the change of the NO$_x$ flux is dominated by chemical loss here. The addition of sinks (Fig. 1a) compensates for such negative divergences, resulting in the emission pattern $E$ (Fig. 1c). Two
point sources emissions on Long Island stand out: Glenwood Landing power station (labelled as power plant PP) and John F. Kennedy (JFK) airport, respectively.

We apply 2D MISATEAM to US cities with populations exceeding 200,000, a categorization that corresponds to medium to large urban areas as designated in Organization for Economic Co-operation and Development (OECD) countries. Adjacent cities, those situated within 50 km of the most populous city in an urban conglomeration, are treated as a single city cluster.
Cities producing significantly weaker NO$_2$ signals than the surroundings are excluded (see criteria in Text S2 of Supplement). The specified criteria yield a combined count of 52 cities and urban conglomerations (refer to Table S1) that are suitable for the application of 2D MISATEAM. We obtained valid results from 39 of these cities shown in Fig. S2. Cities with invalid results are associated with large fitting errors (see Text S2 of the Supplement).

## 2.3 National Emission Inventory (NEI)

We compare emissions derived in this study with those of the bottom-up NEI 2019 processed by George Mason University (Ma and Tong, 2022). NEI is a comprehensive estimate of emissions of criteria air pollutants and their precursors from point, mobile and area sources, which has been widely used to support urban air quality model simulations. The large point sources in NEI are compiled based on direct stack emissions measurements by continuous emissions monitoring systems (CEMS). For non-point sources, the gridded data is generated using spatial proxies such as roadway-level traffic data for distributing
aggregate emissions to grid cells. Figure 1d displays a map of NEI estimates $E_{NEI}$ at a spatial resolution of 12 km for 2019. NEI has been reported to be biased high by 30-70% in the early years of the 2010s (Choi and Souri, 2015; Dickerson et al., 2019). For example, NEI NO$_x$ estimates for 2011 were potentially overestimated by 51–70% over the Baltimore-Washington region (Anderson et al., 2014), 30–60% in the Southeastern US (Travis et al., 2016), and 30–60% over urban areas of Texas (Souri et al., 2016). NEI NO$_x$ emissions for mobile sources alone were reported to be biased high by 28% (McDonald et al.,
2018). We use the lower bound of the reported bias (30%) as the uncertainty for NEI emissions in this study. The spatial

distribution of our estimated emissions $E$ (Fig. 1c) is generally in good agreement with that of NEI emissions (Fig. 1d), with a correlation coefficient of 0.71. More comparisons between these two inventories will be discussed in Section 3.2.

## 2.4 NU-WRF simulations

We use a regional modeling system, the NASA-Unified Weather Research and Forecasting (NU-WRF; Tao et al., 2013; Peters-Lidard et al., 2015), to provide synthetic tropospheric $NO_2$ VCDs $\Omega_{NU-WRF}$ and wind fields $\overrightarrow{v_{NU-WRF}}$ over the continental US. The domain of the simulation is illustrated in Fig. 2 of Liu et al. (2022). We perform the 2016 NU-WRF simulation at a high horizontal spatial resolution of 4 km × 4 km, comparable to the TROPOMI footprint. The meteorological and chemical initial conditions and their lateral boundaries are obtained from NASA's Modern Era Retrospective-Analysis for Research and Applications version 2 (MERRA-2; Gelaro et al., 2017) and the Community Atmosphere Model with chemistry (CAM-chem; Lamarque et al., 2012), respectively. Additional model set-up information, including the chemical mechanism, aerosol module, and emissions are detailed in Tao et al. (2020) and Liu et al. (2022). We integrate simulated $NO_2$ concentrations from the surface to the tropopause to provide $\Omega_{NU-WRF}$. We average wind fields used by NU-WRF from the surface to an altitude of 1000 m to compute $\overrightarrow{v_{NU-WRF}}$.

We apply 2D MISATEAM to the synthetic $NO_2$ VCDs $\Omega_{NU-WRF}$ and wind fields $\overrightarrow{v_{NU-WRF}}$ to map $NO_x$ emissions $E_{NU\text{-}WRF}$ for cities in Table S1. We derive valid results for 33 cities (see Text S2 of Supplement). We have valid results for a smaller number of cities compared to results derived from TROPOMI data, as cities at the edge of the NU-WRF domain, such as Seattle and San Francisco, are omitted. Since the model output partially lacks the data for their inflow/outflow plumes, it fails to satisfy the requirement for the application of 2D MISATEAM. We further compare $E_{NU\text{-}WRF}$ with "true emissions" directly given by the model $E'_{NU-WRF}$ (hereafter referred to as "given emissions"), which are used to drive NU-WRF simulations, to assess the accuracy of 2D MISATEAM.

## 2.5 Performance evaluation

We sum up $NO_x$ emissions from each grid cell within the city's domain to provide a total emission for an individual city. The city domain is defined as 70 km × 70 km around city center for most cities, which is large enough to include all urban areas. We use a larger domain of 100 km × 100 km for New York, Chicago, Los Angeles and Houston due to their larger expanse. The model performance metrics of the Normalized Mean Bias (NMB) and the Root Mean Squared Error (RMSE) for the evaluation are defined as

$$NMB = \frac{\sum_{i=1}^{n}(Emis_i - Emis'_i)}{\sum_{i=1}^{n} Emis'_i} \tag{5}$$

and

$$RMSE = \sqrt{\frac{\sum_{i=1}^{n}(Emis_i - Emis'_i)^2}{n}}, \tag{6}$$

respectively, where *i* denotes an individual city, while *n* refers to the total count of cities utilized for the evaluation. *Emis* denotes the total emission from the MISATEAM-derived datasets (i.e., $E$ or $E_{NU\text{-}WRF}$), and *Emis'* denotes the total emission from benchmark emission datasets (i.e., $E_{NEI}$ or $E'_{NU-WRF}$). We also assess the intracity spatial correlation $R_{intracity}$ by calculating the correlation coefficient of emissions at grid level over the city domain between $E$ and $E_{NEI}$ or between $E_{NU\text{-}WRF}$ and $E'_{NU-WRF}$.

**3 Results and discussions**

**3.1 Validation using NU-WRF simulations**

We compare MISATEAM-derived NO$_x$ emissions $E_{NU\text{-}WRF}$ with given emissions $E'_{NU-WRF}$ to validate 2D MISATEAM. The validation indicates the uncertainty of MISATEAM assuming a best-case scenario with perfect knowledge of the winds and errorless satellite NO$_2$ retrievals. Figure 2 compares the total emissions from the two datasets for the 33 cities (Table S1). The

correlation coefficient is 0.99, which indicates an excellent agreement between the two datasets. The overall bias computed over all valid cities is generally low (NMB = -0.01) and the RMSE is also low (0.21 kg s$^{-1}$).

We assess 2D MISATEAM's performance to infer the spatial distribution of emissions. Figure 3 displays the emission maps around the city of Jacksonville, Florida. $E_{NU\text{-}WRF}$ (Fig 3c) and $E'_{NU-WRF}$ (Fig. 3d) show a high intracity spatial correlation $R_{intracity}$ of 0.92. This good consistency is substantially better than the comparison of $E'_{NU-WRF}$ with the tropospheric NO$_2$ VCD

$\Omega_{NU-WRF}$ (Fig. 3a; $R_{intracity}$ = 0.75), which is often considered to be a reasonable spatial proxy of emissions locations. We further compare the correlation with the comparison of $E'_{NU-WRF}$ with $\Omega_{NU-WRF}$ under calm wind conditions (Fig. 3b). We use the threshold of 2 m s$^{-1}$ as the criterion for calm wind, allowing for the attainment of an adequate sample size. This threshold reduces the error associated with transport by increasing $R_{intracity}$ from 0.75 (Fig. 3a) to 0.80 (Fig. 3b). However, this correlation is still smaller than that between $E_{NU\text{-}WRF}$ and $E'_{NU-WRF}$, suggesting that 2D MISATEAM is successful in allocating emissions

by accounting for NO$_x$ transport. Similar enhanced correlations are observed for all 33 cities. Figure 2 illustrates $R_{intracity}$ of $E_{NU\text{-}WRF}$ and $E'_{NU-WRF}$ for individual cities (0.88 ± 0.06, mean ± standard deviation). These correlations are larger than the comparison of $\Omega_{NU-WRF}$ (0.78 ± 0.09) or $\Omega_{NU-WRF}$ under calm wind conditions (0.80 ± 0.08) against $E'_{NU-WRF}$.

**3.2 TROPOMI-based NO$_x$ emissions**

We compare TROPOMI-based NO$_x$ emissions $E$ with NEI estimates $E_{NEI}$ for 2019 in Fig. 4. The total emission estimates for

individual cities in the two datasets generally agree well with each other, with a correlation $R$ of 0.90. This level of correlation is comparable to the validation using NU-WRF simulations. The relative difference of the total emission between $E$ and $E_{NEI}$ is within the uncertainty range of $E$ (47%; see Section 3.3) for 31 out of 39 cities. The comparison for all cities shows a bias with NMB of -0.24. The bias is likely associated with uncertainties in the TROPOMI NO$_2$ retrievals, which have been reported

to be biased low by 23% on average (van Geffen et al., 2022). The bias may also arise from the uncertainties in NEI, which has been reported to be biased high by over 30% (Section 2.3).

Figure 5 compares the $NO_x$ emission patterns from TROPOMI $NO_2$ with those reported in the NEI, using Dallas and Tucson as case studies. Consistent with observations in New York (as shown in Fig. 1), TROPOMI-derived emission maps reveal several more pronounced point sources as compared to NEI. Notable emissions from the Dallas/Fort Worth International Airport, Perot Field Fort Worth Alliance Airport, and three major cement factories—TXI, Holcim, and Ash Grove—are distinctly evident (Fig. 5a), whereas these details appear diffused in NEI (Fig. 5b). Similarly, emissions from the Asarco Mission copper mine are clearly discernible in the TROPOMI data (Fig. 5c) but are not as apparent in the NEI data (Fig. 5d). We use $R_{intracity}$ to compare the intracity spatial distribution of emissions for more cities in Fig. 4. We upscale $E$ to the same spatial resolution of $E_{NEI}$ to calculate their $R_{intracity}$ (Fig. S3). $R_{intracity}$ between $E$ and $E_{NEI}$ is 0.57 ± 0.16, which is smaller than that between $E_{NU\text{-}WRF}$ and $E'_{NU-WRF}$ in the evaluation using model data (0.88 ± 0.06; Fig. 2). The generally smaller values of $R_{intracity}$ are likely caused by the uncertainties of both TROPOMI-based and NEI emissions. Compared to $E_{NU\text{-}WRF}$ inferred from perfect $NO_2$ columns and wind fields, the uncertainties of TROPOMI $NO_2$ retrievals (25%; van Geffen et al., 2022) and GEOS FP-IT wind reanalysis (30%; Liu et al., 2022) are propagated into the uncertainties of TROPOMI-based emissions $E$ and may result in incorrect spatial patterns. More details about the uncertainties are discussed in Section 3.3. Uncertainties in $E_{NEI}$ also contribute to the disagreement. NEI uses spatial-distribution proxies, such as maps of population densities or road networks, to allocate country-level emissions from non-point sources onto a grid. This procedure may be associated with biases due to either a spatial mismatch between the locations of emissions and spatial proxies or incorrect emission magnitudes. Some hotspots shown in $E$ are missing from $E_{NEI}$, for instance, JFK airport (Fig.1) and Asarco Mission mine (Fig.5), indicating missing sources or misallocation of sources.

Figure 6 presents the trends of derived $NO_x$ emissions across US cities from 2018 to 2021. A significant dip in emissions was observed in 2020, primarily attributed to the diminished $NO_x$ emissions in response to COVID-related lockdown policies (e.g., Liu et al., 2020). The 2020 emissions saw an average reduction of approximately 25% from 2019, in agreement with the findings of other studies (Goldberg et al., 2020; Miyazaki et al., 2021). Emissions in 2021 experienced a rebound compared to those in 2020; however, they are still lower than the pre-pandemic levels. The slopes of the linear regression lines in Fig. 6 decrease from 0.91 in 2019 to 0.85 in 2021. This decline aligns with the long-term trend of decreasing emissions in the US, primarily driven by the downturn trend in vehicular $NO_x$ emissions (McDonald et al., 2018). The reduced slope in 2021 (0.85) relative to 2019 (0.91) may also encapsulate lingering impacts of diminished traffic from the COVID-19 pandemic, since the traffic and commercial flight metrics at the end of 2020 were still substantially lower than their January 2020 levels (Laughner et al., 2021).

### 3.3 Uncertainty analysis

We follow the method proposed by Liu et al. (2022) to evaluate the uncertainties associated with the derived results. We attribute an uncertainty of 20% to the inferred emissions, based on the relative differences between $E_{NU\text{-}WRF}$ and $E'_{NU-WRF}$ for

all 33 cities using the NU-WRF synthetic data (2 ± 24%). Since the differences are less than 20% for most cities (73%), this estimate may be conservative.

Instead of using data spanning multiple years (2018-2021), we apply 1D MISATEAM to annual data to investigate the uncertainty introduced by presuming a consistent $NO_x$ lifetime over several years. For each city, we infer $\tau$ by applying 1D MISATEAM to TROPOMI $NO_2$ VCDs, averaged from May to September for each individual year from 2018 to 2021. Among the cities analyzed, 14 cities (listed in Table S1) have valid $NO_x$ lifetimes available for all four years. The ratio of the standard deviation of the fitted $\tau$ for these individual years to the average $\tau$, as derived from data from 2018 to 2021, stands at 16%. This indicates an uncertainty of 16% in relation to the assumption of a constant $NO_x$ lifetime.

We identify additional uncertainties that may arise when implementing 2D MISATEAM to TROPOMI and GEOS-IT data as opposed to synthetic data, as follows:

- The uncertainty of TROPOMI $NO_2$ observations carries over into the uncertainty of the inferred emissions. We consider an overall uncertainty of 25% for TROPOMI tropospheric $NO_2$ VCDs, following the recommendation from a recent validation using ground-based measurements (van Geffen et al., 2022). The uncertainty originates from various factors, including the spectral fitting process during the retrieval, the separation of stratospheric and tropospheric columns, and the tropospheric air mass factor (AMF).

  AMF accounts for the variable sensitivity of satellite observations to $NO_2$ at different atmospheric altitudes, which is informed by a priori knowledge of $NO_2$ vertical distribution as provided by chemical transport models. Laughner et al. (2016) demonstrated that urban $NO_x$ emissions estimated via $NO_2$ VCDs with daily, high-resolution a priori profiles are considerably higher than those derived from retrievals using coarser resolution profiles. This presents a relevant challenge for our study. To refine the assessment of AMF influences on 2D MISATEAM-derived emissions, a sensitivity analysis where TROPOMI's AMFs are applied to NU-WRF profiles is of help. This would generate $NO_2$ columns with AMF biases, which, when used to calculate emissions, can be contrasted with those derived from idealized columns. Such a comparison would reveal the extent to which AMF biases systematically affect the emissions determined by 2D MISATEAM. Given the extensive computational demands of such a sensitivity analysis, we had to exclude it from this study. Instead, we aim to perform a comprehensive investigation in future research.

  The potential bias arising from the separation of stratospheric and tropospheric columns is eliminated by the employment of the background terms $b$ and $b_{calm}$ within the model functions of MISATEAM. Since the random uncertainty of the tropospheric $NO_2$ observations could be suppressed due to the consideration of long-term means, this estimate may be conservative.

- The presence of clouds is an additional source of uncertainties. We exclude TROPOMI $NO_2$ data with cloud radiance fraction of 0.5 or greater in our analysis. This exclusion may lead to a bias in the $NO_2$ VCD averages, a consequence of eliminating data with changing $NO_x$ lifetime and $NO_x/NO_2$ ratio during cloudy conditions (Geddes et al., 2012). We attribute an uncertainty of 10% to cloud selection criteria based on an assessment carried out at urban locations (Geddes et al., 2012). More sensitivity analysis regarding the impact of clouds has been discussed by Liu et al. (2022).

- The precision of wind fields plays an important role in determining the total uncertainty, as it limits the model functions' ability to describe $NO_2$ transport. We estimate the corresponding uncertainties to be 30% based on an assessment of reanalysis wind products using sounding measurements (refer to Table S3 in Liu et al., 2016a).

We define the overall uncertainty of the inferred emissions as the root of the quadratic sum of the above-mentioned uncertainties, which are assumed to be independent. We thus calculate that the total uncertainty of MISATEAM-derived $NO_x$ emissions for a mid-size US city is 47%.

## 4 Conclusions

In this study, we developed a new method by coupling the 1D CTM-independent methodology (Liu et al. 2022), MISATEAM, with the 2D divergence approach (Beirle et al., 2019) to generate maps of $NO_x$ emissions across US cities, using TROPOMI $NO_2$ observations. This coupled method, 2D MISATEAM, is suitable for sources within polluted backgrounds. Our initial application of the approach used synthetic NU-WRF-generated tropospheric $NO_2$ VCDs over the continental US to evaluate the method. The evaluation yielded robust agreement with the NU-WRF given values, presenting a high correlation coefficient (R = 0.99) and a minimal bias (NMB = -0.01).

Subsequently, we applied 2D MISATEAM to TROPOMI $NO_2$ retrievals across the US cities from 2018 to 2021. We estimated $NO_x$ emissions for 39 US cities. Our resulting total city emissions estimates align strongly with NEI (R = 0.90), albeit with a moderate bias (NMB = -0.24). Our derived emissions delineate differences in spatial patterns across certain cities, implying potential inaccuracies in emission allocation and/or missing sources in NEI. Our current estimates suggest that uncertainties in $NO_x$ emissions arising from 2D MISATEAM method itself are approximately 20% for a large and mid-size US city. Additional uncertainties stem primarily from errors in the reanalysis wind dataset as well as the TROPOMI $NO_2$ retrieval, increasing the overall uncertainties of resulting emissions to about 47%.

2D MISATEAM is contingent on 1D MISATEAM for $NO_x$ lifetime estimation. 1D MISATEAM relies on $NO_2$ observations under calm wind conditions to infer lifetimes, which in turn influences the temporal resolution of the emissions data we can confidently derive. Our investigation indicates that typically 3 to 6 months of TROPOMI data are required to ensure comprehensive data coverage of calm-wind $NO_2$ observations for urban emissions analysis. Therefore, we advise caution when considering the use of 2D MISATEAM for emission estimations over periods shorter than three months, unless we want to use prescribed or externally sourced lifetimes.

In our future research, we plan to extend the application of 2D MISATEAM to observations from geostationary satellites to estimate diurnal variations in urban emissions. Geostationary sensors include the Korean Geostationary Environmental Monitoring Spectrometer (GEMS; Kim et al., 2012), NASA's Tropospheric Emissions: Monitoring of Pollution (TEMPO; Chance et al., 2012), and ESA's Sentinel-4 (Ingmann et al., 2012). These instruments possess spatial resolutions similar to TROPOMI and the validation of NU-WRF simulation (4 km) utilized in this study. For applications based on geostationary satellites with local observation time extending beyond the early afternoon time frame of TROPOMI in this study, further

exploration into the impact of the diurnal cycle of $NO_x$ lifetime will be necessary. When applying the 2D MISATEAM to cities globally, particularly in non-OECD countries, it may be necessary to adjust population thresholds to reflect local demographics and urban profiles. We advise removing cities with too weak emissions signals following this study, i.e., $b_{calm}$/mean VCDs > 50% (Text S2 of the Supplement), as such a high ratio can introduce significant uncertainties associated with determined background $b_{calm}$.

In the next phase of our work, we will strive to harmonize bottom-up and satellite-derived urban emissions estimates to produce a fused emission inventory (Liu et al., 2018). This will enable the provision of timely $NO_x$ emissions estimates that will be of value to both air quality and climate modelling communities.

*Data availability.* The NU-WRF model outputs are available upon request from Zhining Tao (zhining.tao@nasa.gov). Additional data related to this paper may be requested from the corresponding author.

*Author contributions.* Conceptualization and methodology: F.L., J.J., and S.B.; Model simulation: Z.T.; Satellite data processing: S.C.; Wind data processing: Z.F.; NEI data processing: D.T. and S.M.; Formal analysis: F.L.; Visualization: F.L.; Writing—original draft: F.L., review and editing: All authors; Funding acquisition: F.L. and J.J.

*Competing interests.* The authors declare that they have no competing interests.

*Acknowledgements.* This work was funded by NASA through the Aura project data analysis program, the Atmospheric Composition Modeling and Analysis Program (ACMAP) program (grant no. 80NSSC19K0980), and Modeling, Analysis, and Prediction (MAP) Program. We thank Prof. Russell R. Dickerson for helpful discussion on uncertainties in NEI. We thank Dr. Laughner and one anonymous reviewer for helpful comments.

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

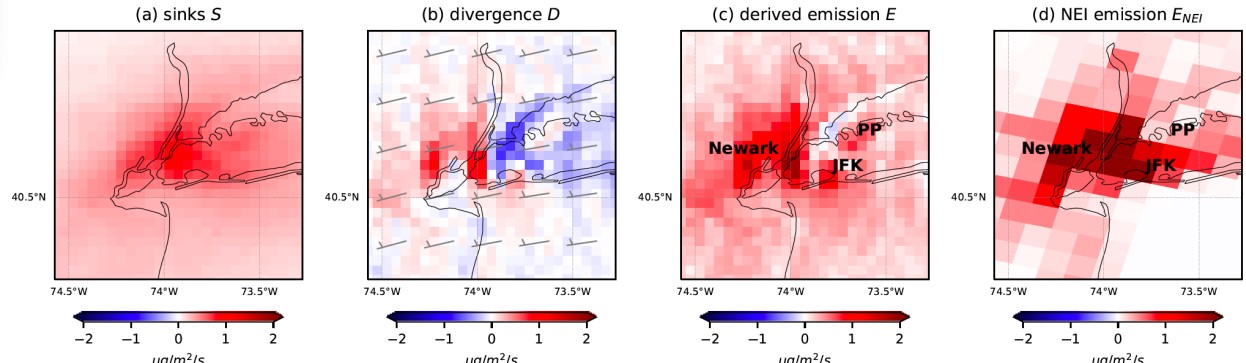

**Figure 1: NOₓ budget inferred from TROPOMI NO₂ observations around New York City from May through September, 2019. (a) sinks $S$, (b) divergence $D$, (c) derived NOₓ emissions $E=S+D$, (d) NEI NOₓ emissions $E_{NEI}$. The locations of Glenwood Landing power station, JFK airport, and the city center of Newark are labeled as PP, JFK, and Newark, respectively. Wind barbs at TROPOMI overpass time from May to September of 2019 are averaged and shown in (b). Wind speed is given in the units of knots, which is a nautical miles per hour (1.9 km per hour). Each short and long barb represents 5 knots (9.3 km/h) and 10 knots (18.5 km/h), respectively.**

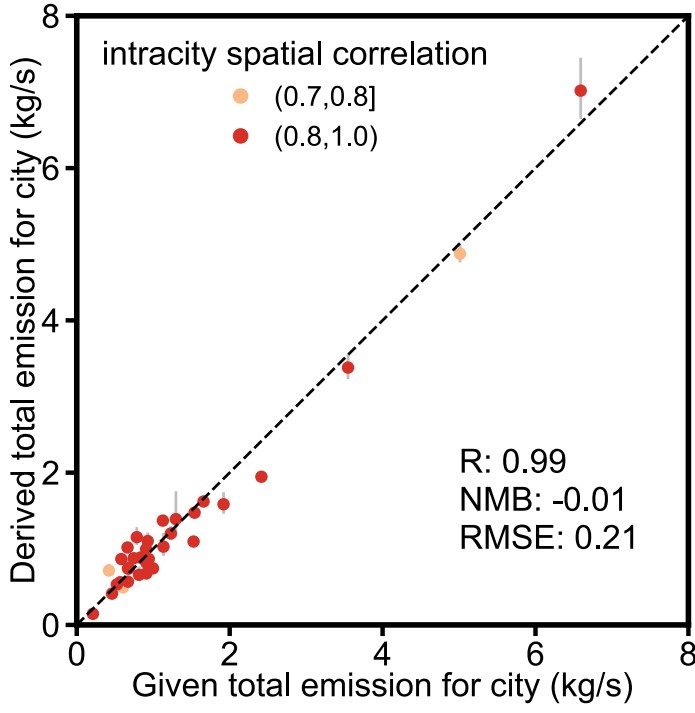

**Figure 2: Scatterplot of the derived NOₓ total emissions for the investigated cities based on the NO₂ tropospheric VCDs simulated by NU-WRF (y axis) as compared to the given emissions used to drive the NU-WRF simulation (x axis). NOₓ emissions from all grid cells within the domain of 70 km × 70 km around city center are summed up to derive the total emission for most cities; a 100 km × 100 km domain is used for New York, Chicago, Los Angeles and Houston. Error bars show the standard error of the**

derived emissions for all wind directions with derived NO$_x$ lifetime $\tau$. Standard error is defined as standard deviation divided by $\sqrt{n}$, with n being the number of wind directions with derived NO$_x$ lifetime $\tau$. The intracity spatial correlation $R_{intracity}$ between the derived and given emissions for individual cities are color coded. The dashed line represents the 1:1 line. Statistics are provided in the inset table.

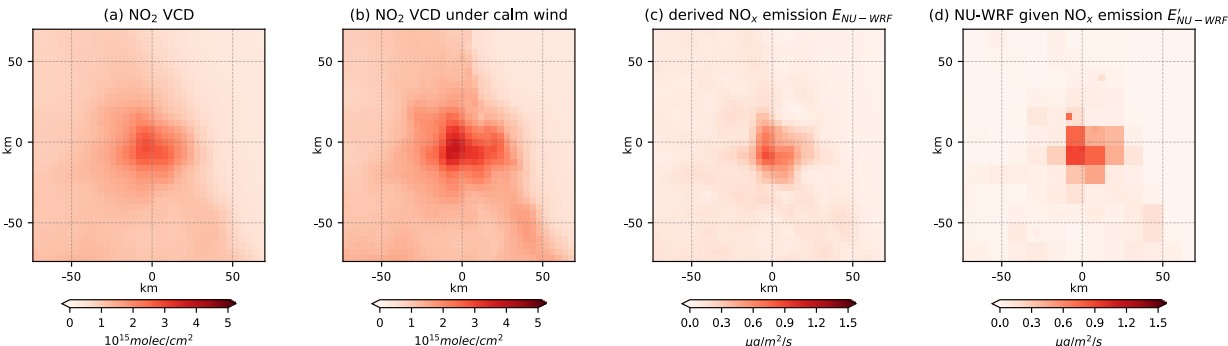

**Figure 3: Improved spatial correlation of derived and given NO$_x$ emissions compared to that of derived NO$_x$ emissions and NO$_2$ columns. (a) Mean NU-WRF tropospheric NO$_2$ VCDs $\Omega_{NU-WRF}$; (b) Mean NU-WRF tropospheric NO$_2$ VCDs under calm wind conditions only; (c) Mean NO$_x$ emission rates $E_{NU\text{-}WRF}$ derived from (a); (d) Mean NO$_x$ emission rates used to drive the NU-WRF simulation $E'_{NU-WRF}$. Hourly mean data at 14:00 LT are averaged from May through September 2016. The city of Jacksonville, Florida is in the centre of the domain shown.**

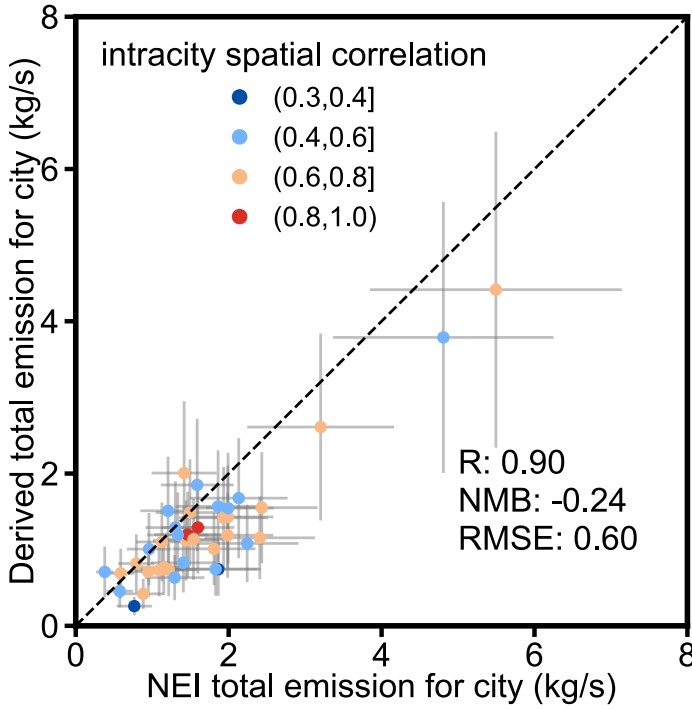

**Figure 4: Similar to Figure 2, but for the comparison between the derived NO$_x$ total emissions based on TROPOMI tropospheric NO$_2$ VCDs (y axis) with NEI total emissions (x axis) for 2019. Error bars show the uncertainties of NEI (30%) and fitted (47%) emissions.**

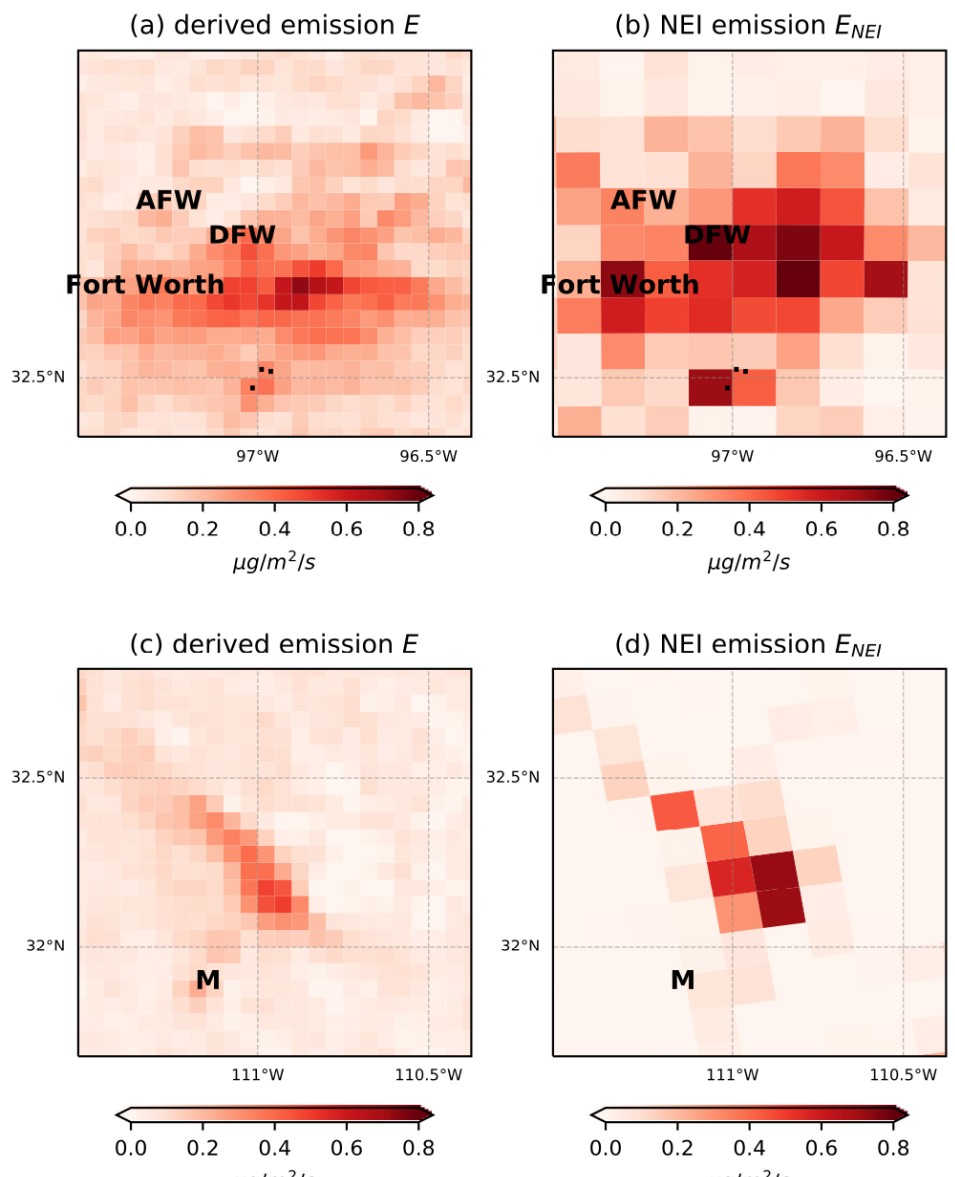

**Figure 5: Comparison of NO$_x$ emissions inferred from TROPOMI NO$_2$ observations with NEI emissions from May through September, 2019. Dallas: (a) TROPOMI-derived NO$_x$ emissions $E$, (b) NEI NO$_x$ emissions $E_{NEI}$. The locations of Dallas/Fort Worth International Airport, Perot Field Fort Worth Alliance Airport and the city center of Fort Worth are labeled as DFW, AFW, and Fort Worth, respectively. The locations of three large cement factories (TXI, Holcim, and Ash Grove) are labeled as dots. Tucson: (c) TROPOMI-derived NO$_x$ emissions $E$, (d) NEI NO$_x$ emissions $E_{NEI}$. The location of a large copper mine Asarco Mission is labeled as M.**

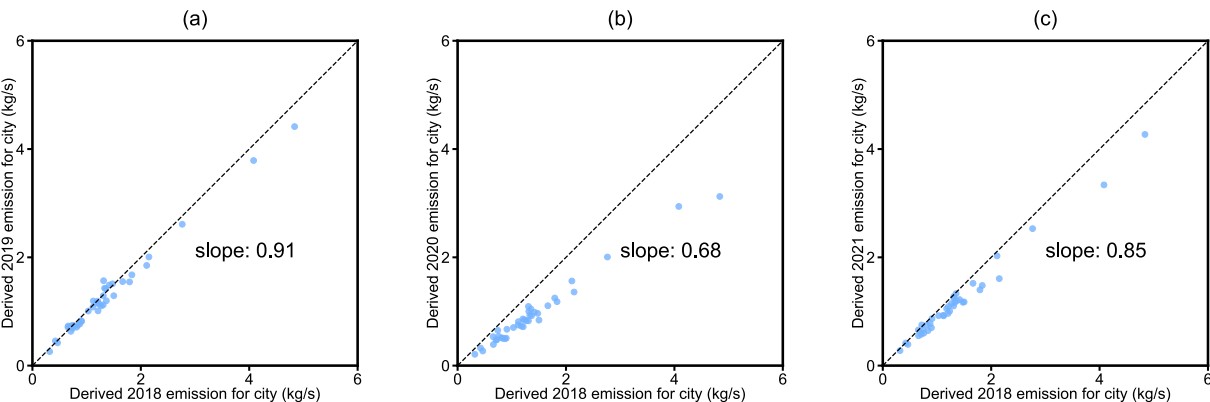

**Figure 6: Comparison of TROPOMI-derived NO$_x$ emission estimates for (a) 2019, (b) 2020, (c) 2021 with those for 2018. The dashed line represents the 1:1 line. The slope of the least-squares linear regression line is provided in the figure.**