# Peer review of "High-resolution Mapping of Nitrogen Oxide Emissions in Large US Cities from TROPOMI Retrievals of Tropospheric Nitrogen Dioxide Columns"

_EGUsphere, 2023_

## Author Comment (AC1)

Below we reply to the reviewer comments point by point. The reviewer comments are shown in *italic*, and corresponding modifications and citations of the manuscript are quoted.

Referee #1

*Liu et al. describes a new approach to inferring $NO_x$ emissions from cities by combining two previously published methods. The first, the 1D MISATEAM approach described in the Liu et al. 2022 reference, is a whole-city mass balance approach that divides space-based $NO_2$ column data by wind speed and direction and finds the emissions that balance the transport and chemical removal of $NO_2$. The second, the divergence-based approach described in the Beirle et al. 2019 reference, is also a mass balance approach in essence, but one which is applied to individual grid cells. There the difference between horizontal flux of $NO_2$ into and out of a grid cell is taken to represent the sum of emissions and sinks in that grid cell, with the sink assumed to be the first-order chemical loss of $NO_2$. By combining these two methods, this paper is able to use lifetimes and background $NO_2$ columns derived from the whole-city analysis in the grid-cell level calculation.*

*This is an interesting evolution of our ability to directly constrain $NO_x$ emissions from space without use of computationally expensive models. The paper generally does a good job of evaluating the veracity of this method using synthetic data, which demonstrates that this method has good skill in recovering known emissions assuming no systematic biases. The uncertainty estimates are reasonable and justified, though I have one suggestion for an additional test. There are a few points that can be strengthened, which I will detail below. However, this is already a strong paper and I recommend publication after the points below are addressed.*

**Response:** We thank Dr. Laughner for the encouraging comments. All comments and suggestions have been considered carefully and addressed below.

*- Point 1: the only limitation I saw in the validation with NU-WRF data was that possible systematic biases in the AMF were not tested. If I understood correctly, the synthetic $NO_2$ columns used in the validation were an integration of the NU-WRF profiles without any AKs from the $NO_2$ retrieval applied. Thus, this essentially assumes perfect AMFs. We know from Laughner et al. 2016 (https://doi.org/10.5194/acp-16-15247-2016) that AMF biases from the a priori profiles can lead to biases in the emissions and lifetime derived from methods similar to*

*the 1D MISATEAM approach. I suspect that such biases would be fairly small in this case, as the MINDS NO$_2$ retrieval used in this study does have reasonably high resolution a priori profiles (0.25 deg). But we also know from Valin et al. 2011 (https://doi.org/10.5194/acp-11-11647-2011) that even at ~25 km, chemical transport models don't capture the full nonlinearity of NO$_x$ chemistry.*

*I think that there is a straightforward way to test whether any AMF biases present in the NO$_2$ retrieval are large enough to affect the 2D MISATEAM method. If you were to repeat the test where you derived emission by applying 2D MISATEAM to the synthetic NU-WRF columns, but this time apply MINDS AKs to the NU-WRF profiles rather than doing a simple column integration, then the emissions derived in this test should reflect the impact of an imperfect AMF. By comparing these imperfect AMF emissions against the emissions derived using the NU-WRF columns without AKs (that represent a "perfect" AMF case), that difference should reveal any systematic impact of systematic AMF biases on the 2D MISATEAM emissions.*

**Response:** We acknowledge the limitation of using integrated NU-WRF columns as synthetic satellite observations, specifically the uncertainties arising from AMF errors. The recommendation to apply MINDS AKs to NU-WRF profiles to probe AMF impacts is well-received. While such an in-depth analysis aligns with the insights provided by Laughner et al. (2016) and would be valuable, our current study's focus is to present the optimal performance of 2D MISATEAM using "perfect columns". Given the extensive computational demands of such sensitivity analyses, we are going to perform a comprehensive investigation into the AMF's impact on 2D emissions in future study, following Laughner et al. (2016). We elaborate on the limitation in the revised Section 3.3, as follows:

"AMF accounts for the variable sensitivity of satellite observations to NO$_2$ at different atmospheric altitudes, which is informed by a priori knowledge of NO$_2$ vertical distribution as provided by chemical transport models. Laughner et al. (2016) demonstrated that urban NO$_x$ emissions estimated via NO$_2$ VCDs with daily, high-resolution a priori profiles are considerably higher than those derived from retrievals using coarser resolution profiles. This presents a relevant challenge for our study. To refine the assessment of AMF influences on 2D MISATEAM-derived emissions, future work could include a sensitivity analysis where TROPOMI's AMFs are applied to NU-WRF profiles. This would generate NO$_2$ columns with AMF biases, which, when used to calculate emissions, can be contrasted with those derived from

idealized columns. Such a comparison would reveal the extent to which AMF biases systematically affect the emissions determined by 2D MISATEAM.”

*- Point 2: there is one sentence at the end of Sect. 3.2 that could use additional justification - "The slopes of the linear regression lines in Fig. 5 decrease from 0.91 in 2019 to 0.85 in 2021. This can be attributed to the long-term trend of decreasing emissions in the US, primarily driven by the downturn trend in vehicular $NO_x$ emissions (McDonald et al., 2018)." My concern is that 2021 may still include effects from the COVID-19 pandemic. In Laughner et al. 2021 (https://doi.org/10.1073/pnas.2109481118), we see that metrics for traffic and commercial flights (globally as well as in Los Angeles and San Francisco specifically), remain well below their Jan 2020 levels at the end of 2020.*
*If this conclusion (that the 2018 to 2021 decrease in $NO_x$ emissions is part of the long term trend in the US) is an important part of your work, I'd strongly suggest looking at at least the Google mobility trends (https://www.google.com/covid19/mobility/) and possibly state/city level traffic metrics (e.g. CalTrans PEMS, https://pems.dot.ca.gov/) to check if the underlying traffic driving a substantial part of these emissions had returned to pre-pandemic levels to support this conclusion. If this conclusion isn't critical, then I would recommend adding a caveat that it could include some lingering effects of reduced traffic during the pandemic.*

**Response:** We concur with your observation that the observed decrease in $NO_x$ emissions from 2018 to 2021 was not solely caused by the long-term trend in the US. Since the emissions reduction from 2018 to 2021 is not our major conclusion in this study, we have added a caveat in Sect. 3.2 to underscore the possibility that the decrease in $NO_x$ emissions during the mentioned period might be influenced by the lingering effects of reduced traffic due to the pandemic, in addition to the long-term trend, as follows:

“The slopes of the linear regression lines in Fig. 5 decrease from 0.91 in 2019 to 0.85 in 2021. This decline aligns with the long-term trend of decreasing emissions in the US, primarily driven by the downturn trend in vehicular $NO_x$ emissions (McDonald et al., 2018). The reduced slope in 2021 (0.85) relative to 2019 (0.91) may also encapsulate lingering impacts of diminished traffic from the COVID-19 pandemic, since the traffic and commercial flight metrics at the end of 2020 were still substantially lower than their January 2020 levels (Laughner et al., 2021).”

*- Point 3: unless I misunderstood, it seems like you should be able to check for closure of emissions between the 1D and 2D MISATEAM results. That is, the emissions which could be output by the 1D MISATEAM algorithm as in Liu et al. 2022 should represent the total city emissions, and so should be approximately equal to the sum of the gridded emissions derived in the 2D MISATEAM approach. In particular, I wonder if this could be a useful quality check to allow you to expect this method to more cities around the world without needing to validate each city with synthetic NU-WRF data. It would be interesting to see if the cities listed in Table S1 that failed NU-WRF validation also have these two emission estimates (from 1D MISATEAM and this method) differ by more than their uncertainty.*

**Response:** We confirm that the 1D and 2D MISATEAM indeed achieve closure. Figure below compares the emissions estimates from both methods. They show a small relative difference of - 5% on average. However, while this consistency might suggest a potential quality check, it cannot be used as such. The reason being that the observed consistency across all sources is inherent to both methods, as they are fundamentally based on the mass balance principle. In this way, the cities listed in Table S1 that failed NU-WRF validation demonstrate a similar consistency between the two methods, with a relative difference $1\% \pm 14\%$ (mean $\pm$ STD).

[Figure]

**Figure: Scatterplot of the derived $NO_x$ total emissions for the investigated cities based on the $NO_2$ tropospheric VCDs simulated by NU-WRF using 2D (y axis) as compared to those using 1D (x axis) methods. $NO_x$ emissions from all grid cells within the domain of 70 km × 70 km around city center are summed up to derive the total emission for most cities; a 100 km × 100 km domain is used for New York, Chicago, Los Angeles and Houston.**

We clarify this in Section 2.2, as follows:

"1D and 2D MISATEAM methods deliver consistent estimates for total emissions, displaying a small relative difference of -5% ± 9%. While this result demonstrates the internal consistency of the MISATEAM methods, it's important to note that such coherence is due to the shared fundamental principle of mass balance underlying both methodologies. Therefore, the similarity in emissions estimations should not be viewed as an independent validation metric."

*- Point 4: it seems like the 2D MISATEAM method implicitly assumes that the background $NO_2$ is the $NO_2$ above the boundary layer. Otherwise, it doesn't make sense to me to use only the non-background $NO_2$ in the calculation of chemical loss (Eq. 3). Is this true? If so, it would be good to explicitly state that assumption.*

**Response:** The 2D MISATEAM methodology does not make implicit assumptions regarding the vertical distribution of background $NO_2$. Instead, our approach utilizes non-background $NO_2$ specifically to compute the associated transport and chemical loss, with the primary objective of deriving non-background emissions. We have clarified this in Section 2.2, as follows:

"Note that we subtract the $NO_2$ background $b$ from $\wedge$ in the calculation of the divergences and the sinks, because we aim to remove the natural and non-local contributions from the total emissions for each urban area in order to infer urban emissions."

*- Point 5: I was initially confused by the discussion of the lifetime uncertainty in Sect. 3.3 (lines 221 to 225). The way the uncertainty analysis was presented made me think that the lifetime in Eq. (3) was a single lifetime used for all cities, rather than having unique lifetimes for each city but that does not change in time. On a second read, I found the sentence at line 109 that indicated that the lifetime and background were calculated for each city. Still, it might be good to restate in Sect. 3.3 that the constant lifetime over several years is different for each city. Also, I assume that the reason only 14 cities could be used for the year-by-year lifetime standard deviation in the uncertainty analysis is that they were the only cities with enough good quality data to derive robust lifetimes separately for each year? If so, please state that and list which cities those 14 were. That will be useful documentation in case it is later found that those 14*

*cities aren't representative of the trend in lifetime for the 39 cities for which emissions were estimated.*

**Response:** Following the recommendation, we have clarified in Section 3.3 that the lifetime is inferred individually for each city. Additionally, we have provided an explanation regarding the selection of the 14 cities for the year-by-year lifetime analysis and highlighted these 14 cities in the revised Table S1. The updated Section 3.3 is as follows:

"Instead of using data spanning multiple years (2018-2021), we apply 1D MISATEAM to annual data to investigate the uncertainty introduced by presuming a consistent $NO_x$ lifetime over several years. For each city, we infer $\tau$ by applying 1D MISATEAM to $NO_2$ VCDs, averaged from May to September for each individual year from 2018 to 2021. Among the cities analyzed, 14 cities (listed in Table S1) have valid $NO_x$ lifetimes available for all four years."

---

## Author Comment (AC2)

Below we reply to the reviewer comments point by point. The reviewer comments are shown in *italic*, and corresponding modifications and citations of the manuscript are quoted.

Referee #2

*This manuscript presents an improved top-down $NO_x$ emission estimate methodology using TROPOMI for select US cities and discusses the method validation and outputs. The improved methodology is the combination of two previously published and widely accepted methods, developed respectively by the two leading coauthors of the manuscript. While I like the manuscript is concise and generally well-written, my main concerns are lack of explanation in some key places and also the lack of details on the derived emissions.*

**Response:** We greatly appreciate your insightful review and the positive remarks regarding the conciseness and clarity of our manuscript. We will expand the sections to provide a more in-depth explanation of our methodology and the derived emissions. These enhancements will address the current gaps in explanation and detail as noted in your comments.

1. *The improved emission mapping algorithm, 2D MISATEAM, is the foundation of the paper. I found Section 2.2 as written does not provide a sufficient justification and motivation for it. Line 93-94 simply states that 2D MISATEAM "is capable of mapping NOx emissions over urban areas.". This statement is not followed by any justification, making it a speculation. The authors need to provide more details on the precedent methodologies, namely 1D MISATEAN and 2D divergency method, as to their respective pros and cons that motivate the development of 2D MISATEAM and how the presented 2D MISATEAM method overcomes the shortcomings of the precedent methods.*

**Response:** We appreciate the opportunity to clarify the advancement 2D MISATEAM represents. The 1D approach effectively quantifies total city-level $NO_x$ emissions, treating urban areas as point sources, yet it lacks the ability to provide spatial distribution details. The 2D divergence method improves upon spatial resolution but is dependent on additional sources for $NO_x$ lifetime estimations, either through a constant prescribed lifetime or external data. 2D MISATEAM synergistically combines these two approaches to infer spatial emissions distributions directly and independently, deriving $NO_x$ lifetimes internally from $NO_2$

measurements and winds info without relying on external lifetime data. We update Section 2.2 accordingly as follows:

"We couple our 1D CTM-Independent SATellite-derived Emission estimation Algorithm for Mixed-sources (MISATEAM; Liu et al., 2022) with the 2D divergence method of Beirle et al. (2019). The 1D MISATEAM quantifies the magnitude of city-level $NO_x$ lifetime and emissions by conceptualizing urban areas as point sources and thus does not capture the spatial variability within the urban areas. Conversely, the 2D divergence method allows for the resolution of finer spatial details in $NO_x$ distributions but relies on additional, often external, sources for determining $NO_x$ lifetimes, which can be a significant limitation. The coupled algorithm (hereafter referred to as 2D MISATEAM for simplicity) leverages the strengths of both: it maps $NO_x$ emissions over urban areas with enhanced spatial detail and does so independently by deriving lifetimes directly from $NO_2$ measurements, thereby overcoming the need for prescribed or externally sourced lifetime constraints."

2. *The limitation of 2D MISATEAM and its applicability to outside of US should be discussed better. The paper uses the new methodology to large US cities based on population. If the community wants to adopt 2D MISATEAM to other countries/regions which have different population sizes than the US, what should they use to select the suitable places? Does it require that the city has a well-defined urban core with concentrated emissions so that it means certain assumptions in the shape of the urban plumes, etc?*

**Response:** Urban $NO_x$ emissions are closely linked to population size, as densely populated areas often have higher fossil fuel consumption, leading to greater $NO_x$ emissions. We thus choose the populations exceeding 200,000 to select US cities, as it is a categorization that corresponds to medium to large urban areas as designated in Organization for Economic Co-operation and Development (OECD) countries. For non-OECD countries, local demographic and urban characteristics may necessitate different population thresholds. However, the application of the 2D MISATEAM is not stringently tied to this population parameter. Following this study, we recommend excluding cities where the background to mean VCD ratio is too high (above 50%) to limit the uncertainties associated with background determination. The urban area's size or the specific shape of the urban plumes are not essential to MISATEAM. We clarify this in Section 4 as follows:

"When applying the 2D MISATEAM to cities globally, particularly in non-OECD countries, it may be necessary to adjust population thresholds to reflect local demographics and urban profiles. We advise removing cities with too weak emissions signals, i.e., $b_{calm}$/mean VCDs > 50% (Text S2 of the Supplement), as such a high ratio can introduce significant uncertainties associated with determined background $b_{calm}$."

3.  *The paper states that TROPOMI $NO_2$ columns from May – Sep of each year during 2018-2021 were used to derive top-down emissions. It is not clear what's the temporal time step of 2D MISATEAM when it derives top-down emissions. Does it apply to monthly-averaged TROPOMI NO2 using monthly-averaged winds to derive monthly mean emissions per city, or does it apply to May-Sep mean of those quantities and estimate May-Sep averaged emissions? Also, what determines the temporal resolution suitable for 2D MISATEAM? For example, if one wants to use to derive weekly or even daily emissions, assuming TROPOMI has plenty of good pixels for such a short time period, is there anything assumed within 2D MISATEAM that prevents such application from being successful?*

    **Response:** The 2D MISATEAM has been applied to May-Sep mean of $NO_2$ columns and wind data to estimate May-Sep averaged emissions. We have clarified this in Section 2.2 as follows:

    "We average both $NO_2$ VCDs and reanalysis wind data from May to September each year. We then use those averaged data to infer $NO_x$ emissions $E$ by summing the divergence of the $NO_x$ flux $D$ with the $NO_x$ sink $S$ based on the continuity equation for steady state."

    2D MISATEAM is contingent on 1D MISATEAM for $NO_x$ lifetime estimation, which is a determining factor for the temporal resolution of derived emissions.1D MISATEAM relies on $NO_2$ observations under calm wind conditions to infer lifetimes. We perform a sensitivity analysis for all US cities investigated in this study, using 1 to 5 months of NU-WRF data. we have identified trends in data coverage. The resulting data, as detailed in the subsequent table, show that a one-month period provides complete data coverage for less than 60% of cities. Coverage improves to approximately 80% with a three-month data span and does not significantly increase with longer data spans. Given that actual satellite observations necessitate cloud filtering, which further reduces data availability, we generally observe that a 3-6 month period of TROPOMI observations is needed to ensure sufficient coverage over urban areas. Therefore, we do not advise the use of 2D MISATEAM for emission estimations over periods

shorter than 3 months if we do not want to use prescribed or externally sourced lifetimes. Nonetheless, should lifetime estimates be obtained from external sources, 2D MISATEAM could theoretically be adapted to calculate $NO_x$ emissions over shorter intervals.

Table Perecentage of cities with complete data coverage of $NO_2$ VCDs under calm-wind situations over the urban areas.

| length of data used for averaging | percentage of cities |
|---|---|
| 1 month | 58% |
| 2 month | 68% |
| 3 month | 77% |
| 4 month | 80% |
| 5 month | 80% |

We clarified this limit in the conclusion, as follows:

"2D MISATEAM is contingent on 1D MISATEAM for $NO_x$ lifetime estimation. 1D MISATEAM relies on $NO_2$ observations under calm wind conditions to infer lifetimes, which in turn influences the temporal resolution of the emissions data we can confidently derive. Our investigation indicates that typically 3 to 6 months of TROPOMI data are required to ensure comprehensive data coverage of calm-wind $NO_2$ observations for urban emissions analysis. Therefore, we advise caution when considering the use of 2D MISATEAM for emission estimations over periods shorter than three months, unless we want to use prescribed or externally sourced lifetimes."

4. *The manuscript does not provide a good amount of details on the derived top-down emissions over the US cities and how they compare to the NEI inventory. In the abstract, the last sentence states that "there are noticeable differences in the spatial patterns of emissions in some cities" between the TROPOMI-derived and NEI inventory. I don't find where the manuscript elaborated on this main point. Figure 1 is the only place I saw the spatial pattern with a city is presented, but that's only for NYC. To prove the point for "some cities", at least two more cities should be presented. Is the within-city spatial pattern resolution a key strength of 2D MISATEAM? What's the key innovation in it that makes it outperform 1D MISATEAM and 2D divergency method in achieving this?*

**Response:** We add a new figure (Figure 5) to compare emission maps between the TROPOMI-derived and NEI inventory for two more cities, Dallas and Tucson. We also use the intracity spatial correlation $R_{intracity}$, the correlation coefficient of emissions at grid level over the city domain between TROPOMI-derived emissions $E$ and NEI emissions $E_{NEI}$, to compare the spatial patterns of both inventories in Figure 4.

We extend the discuss in Section 3.2 as follows:

"Figure 5 compares the $NO_x$ emission patterns from TROPOMI $NO_2$ with those reported in the NEI, using Dallas and Tucson as case studies. Consistent with observations in New York (as shown in Fig. 1), TROPOMI-derived emission maps reveal several more pronounced point sources as compared to NEI. Notable emissions from the Dallas/Fort Worth International Airport, Perot Field Fort Worth Alliance Airport, and three major cement factories—TXI, Holcim, and Ash Grove—are distinctly evident (Fig. 5a), whereas these details appear diffused in NEI (Fig. 5b). Similarly, emissions from the Asarco Mission copper mine are clearly discernible in the TROPOMI data (Fig. 5c) but are not as apparent in the NEI data (Fig. 5d).

We use $R_{intracity}$ to compare the intracity spatial distribution of emissions for more cities in Fig. 4. We upscale $E$ to the same spatial resolution of $E_{NEI}$ to calculate their $R_{intracity}$ (Fig. S3). $R_{intracity}$ between $E$ and $E_{NEI}$ is $0.57 \pm 0.16$, which is smaller than that between $E_{NU\text{-}WRF}$ and $E'_{NU-WRF}$ in the evaluation using model data ($0.88 \pm 0.06$; Fig. 2). The generally smaller values of $R_{intracity}$ are likely caused by the uncertainties of both TROPOMI-based and NEI emissions. Compared to $E_{NU\text{-}WRF}$ inferred from perfect $NO_2$ columns and wind fields, the uncertainties of TROPOMI $NO_2$ retrievals (25%; van Geffen et al., 2022) and GEOS FP-IT wind reanalysis (30%; Liu et al., 2022) are propagated into the uncertainties of TROPOMI-based emissions $E$ and may result in incorrect spatial patterns. More details about the uncertainties are discussed in Section 3.3. Uncertainties in $E_{NEI}$ also contribute to the disagreement. NEI uses spatial-distribution proxies, such as maps of population densities or road networks, to allocate country-level emissions from non-point sources onto a grid. This procedure may be associated with biases due to either a spatial mismatch between the locations of emissions and spatial proxies or incorrect emission magnitudes. Some hotspots shown in $E$ are missing from $E_{NEI}$, for instance, JFK airport (Fig.1) and Asarco Mission mine (Fig.5), indicating missing sources or misallocation of sources."

The capacity for within-city spatial pattern resolution is indeed a principal advantage of all 2D methodologies, including both the 2D MISATEAM and divergence methods. The key innovation

of the 2D MISATEAM, which enables it to surpass the 1D MISATEAM in this regard, has been detailed previously in our response to your Comment 1.

*Minor comments:*
- *Line 73: the TROPOMI footprint changes over the study period. Specify the changes.*

  **Response:** We specify it in the revised manuscript as follows:

  "It has a ground pixel size at nadir of 7.5 km×3.5 km before August 6, 2019, and improved to 5.5 km ×3.5 km afterwards".

- *Line 79: Specify what official product of TROPOMI NO₂ is and provides a reference.*

  **Response:** We specify it in the revised manuscript as follows:

  "We selected TROPOMI $NO_2$ retrieved by NASA Goddard Space Flight Center (GSFC), TROPOMI Multi-Decadal Nitrogen Dioxide and Derived Products from Satellites (MINDS) $NO_2$ product (Lamsal et al., 2022), in this study."

- *Line 88: There is no cloud screening applied? Why?*

  **Response:** We use the TROPOMI quality assurance value filter (qa_value > 0.75) to remove low-quality observations. This filter removes cloud-covered scenes with cloud radiance fraction > 0.5. We add this in the revised manuscript as follows:

  "Only high-quality pixels with a quality assurance value (qa_value) above 0.75 are considered for averaging, which excludes cloud-covered scenes with cloud radiance fraction > 0.5."

- *Line 245: The uncertainty in the derived NOx emissions is 47%. How does this affect the comparison with NEI? The abstract last sentence attributed all the discrepancy to the NEI. Will the uncertainty in the top-down emissions explain some of the discrepancies?*

  **Response:** We have discussed the impact of the uncertainties in the TROPOMI-derived emissions on the comparison with NEI in Section 3.2, as follows:

  "We compare TROPOMI-based $NO_x$ emissions $E$ with NEI estimates $E_{NEI}$ for 2019 in Fig. 4…. The relative difference of the total emission between $E$ and $E_{NEI}$ is within the uncertainty range of $E$ (47%; see Section 3.3) for 31 out of 39 cities. The comparison for all cities shows a bias with

NMB of -0.24. The bias is likely associated with uncertainties in the TROPOMI $NO_2$ retrievals, which have been reported to be biased low by 23% on average (van Geffen et al., 2022).

The comparison of intracity spatial distribution of emissions $R_{intracity}$ shows more disparity in Fig. 4. $R_{intracity}$ between $E$ and $E_{NEI}$ is $0.57 \pm 0.16$, which is smaller than that between $E_{NU\text{-}WRF}$ and $E'_{NU-WRF}$ in the evaluation using model data ($0.88 \pm 0.06$; Fig. 2). The generally smaller values of $R_{intracity}$ are likely caused by the uncertainties of both TROPOMI-based and NEI emissions. Compared to $E_{NU\text{-}WRF}$ inferred from perfect $NO_2$ columns and wind fields, the uncertainties of TROPOMI $NO_2$ retrievals (25%; van Geffen et al., 2022) and GEOS FP-IT wind reanalysis (30%; Liu et al., 2022) are propagated into the uncertainties of TROPOMI-based emissions $E$ and may result in incorrect spatial patterns."

We also revise the last sentence of the abstract to reflect the impact of the uncertainties in the TROPOMI-derived emissions, as follows:

"There are noticeable differences in the spatial patterns of emissions in some cities. Our analysis suggests that uncertainties in TROPOMI-based emissions and potential misallocation of emissions and/or missing sources in bottom-up emission inventories both contribute to these differences."

---

## Author Response (AR2)

Below we reply to the reviewer comments point by point. The reviewer comments are shown in *italic*, and corresponding modifications and citations of the manuscript are quoted.

*Thank you to Dr. Liu and all the authors for a clear response to my comments. My concerns have been addressed, I would just suggest that some of the points touched on in their response be included in the paper.*
**Response**: We thank Dr. Laughner for the comments again. All suggested points have been added in the revised manuscript.

*- I'm surprised that applying the TROPOMI AKs is as significant a computational effort as it sounds, but I accept that I'm not familiar with your set up. Perhaps include in the AMF discussion under sect. 3.3 that carrying out this analysis is computationally expensive, which is why it will be done in the future.*
**Response**: We add the statement about future work in Section 3.3, as below:
"Given the extensive computational demands of such a sensitivity analysis, we had to exclude it from this study. Instead, we aim to perform a comprehensive investigation in future research."

*- In the revised sect. 2.2 when discussing the consistency between the 1D and 2D methods, I suggest you include that the cities which failed the validation also show consistency. That's a nice piece of concrete evidence that the two methods are self-consistent by design and that comparing them does not add information.*
**Response**: Following the recommendation, we include that the cities which failed the validation also show consistency in Section 2.2, as follows:
"The cities listed in Table S1 that failed NU-WRF validation demonstrate a similar consistency between the two methods, with a relative difference $1\% \pm 14\%$."

*- For the background $NO_2$, what I was getting at more generally is that to use the background as done here, we have to assume that the background $NO_2$ is constant over each city's plume. Physically, to me this means that either the background $NO_2$ is above the PBL, where all the urban chemistry won't affect it, or that the steady state between non-local emissions, $NO_x$ recycling from gas/aerosol phase reservoirs, and loss processes remains unchanged between the up- and down- wind sectors. Given the plethora of emissions coming from a city, it makes more sense to me that an invariant background would come from free troposphere or residual stratosphere $NO_2$. However, I think a statement in sect. 2.2 that this method assumes a relatively constant background would be sufficient.*
**Response**: We add a statement in Section 2.2 that this method assumes a relatively constant background, as follows:
"Note that we assume a relatively constant background and subtract the $NO_2$ background $b$ from $\Omega$ in the calculation of the divergences and the sinks"